

# Consumption of antimicrobial manuka honey does not significantly perturb the microbiota in the hind gut of mice

Doug Rosendale[1], Christine A. Butts[1], Cloe Erika de Guzman[2], Ian S. Maddox[3], Sheridan Martell[1], Lynn McIntyre[3,4], Margot A. Skinner[5], Hannah Dinnan[1] and Juliet Ansell[1,6]

[1] Food, Nutrition & Health Group, The New Zealand Institute for Plant & Food Research Limited, Palmerston North, New Zealand
[2] Translational Genomics Group, Institute of Health and Biomedical Innovation, Brisbane, Queensland, Australia
[3] Massey University, College of Sciences, Auckland, New Zealand
[4] Department of Food Science and Agri-Food Supply Chain Management, Harper Adams University, Newport, Shropshire, United Kingdom
[5] Food Science, School of Chemical Science, The University of Auckland, Auckland, New Zealand
[6] Zespri International Limited, Mt Maunganui, New Zealand

Corresponding author
Doug Rosendale,
douglas.rosendale@plantandfood.co.nz

## ABSTRACT

The aim of this study was to test the hypothesis that consuming manuka honey, which contains antimicrobial methylglyoxal, may affect the gut microbiota. We undertook a mouse feeding study to investigate whether dietary manuka honey supplementation altered microbial numbers and their production of organic acid products from carbohydrate fermentation, which are markers of gut microbiota function. The caecum of C57BL/6 mice fed a diet supplemented with antimicrobial UMF® 20+ manuka honey at 2.2 g/kg animal did not show any significantly changed concentrations of microbial short chain fatty acids as measured by gas chromatography, except for increased formate and lowered succinate organic acid concentrations, compared to mice fed a control diet. There was no change in succinate-producing Bacteroidetes numbers, or honey-utilising Bifidobacteria, nor any other microbes measured by real time quantitative PCR. These results suggest that, despite the antimicrobial activity of the original honey, consumption of manuka honey only mildly affects substrate metabolism by the gut microbiota.

## INTRODUCTION

New Zealand manuka (*Leptospermum* spp.) honey is renowned for its antimicrobial activity, primarily attributed to methylglyoxal (MGO) (*Kwakman et al., 2010*). MGO alone or in manuka honey is a potent antimicrobial compound *in vitro* (*Ferguson et al., 1998*; *Kwakman et al., 2010*; *Lu et al., 2013*). Consumption of manuka honey may deliver antimicrobial MGO, prebiotic honey oligosaccharides or bioactive phytochemicals to the gut. We do not know if consuming high doses of manuka honey affects gut bacterial populations *in vivo*. The role of the commensal microbiota in human health is becoming

increasingly well documented (*Jones, 2013*). To determine whether daily consumption of high relative doses of manuka honey adversely affects the function of the gut microbiota in healthy individuals, a feeding study was conducted using an established C57BL/6 mouse animal model of gut microbial function (*Shu & Gill, 2002*).

## MATERIALS & METHODS

### Animal feeding study

This animal feeding study was approved by the AgResearch Grasslands Animal Ethics Committee, Palmerston North, NZ, Ethics Application 11163. It was carried out in the Food Evaluation Unit, Plant & Food Research, Palmerston North, under conventional housing conditions with healthy male C57BL/6J mice fed AIN-76A diet and drinking water *ad libitum*.

After 1 week of adaptation (approx. 5–6 weeks old), 40 mice were randomly assigned to either control or treatment groups ($n = 20$) and fed the experimental diets for 28 days. The treatment diet was AIN-76A supplemented with UMF® 20+ manuka honey (Comvita NZ Ltd., Paengaroa, NZ) (15.0 g/kg diet), replacing a part of the sucrose component of the diet. This honey value was chosen based on the amount of honey used elsewhere (*Nasuti et al., 2006*) where, at an estimated 3 g diet consumed/day/animal, would equate to 45 mg manuka honey consumed/day/animal, or with 20 g mice, 2.2 g/kg animal. This same batch of manuka honey has been previously determined to be antimicrobial *in vitro* (*Rosendale et al., 2008*), and possesses an MGO concentration of 610 mg/kg, equating to 8 mM MGO in the honey; 0.12 mM in the diet.

Mice were checked daily and weighed three times per week. Food consumption was recorded weekly throughout the trial. After 28 days of feeding, the mice were killed via $CO_2$ asphyxiation and the caecum, a well-defined anatomic structure at the entrance to the colon harbouring a dense ($10^{11}$–$10^{12}$ cells/mL contents) microbiota (*Rawls et al., 2006*), was removed, snap-frozen in liquid nitrogen, and stored at $-80\,°C$ until use.

### Organic acid analysis

Organic acids from the caecum digesta were extracted into diethyl ether and analysed in duplicate using a Shimadzu gas chromatograph equipped with a flame ionisation detector (GC-FID) (Shimadzu Scientific Instruments Pty Ltd., Australia), as described previously (*Paturi et al., 2010*).

### Microbial quantification

Caecal microbial DNA was extracted and used as a template for absolute microbial quantification by real time (RT)-qPCR using a standard SYBR-green based assay and a Roche Lightcycler 480 (Roche Diagnostics GmbH, Werk Penzberg, Germany) in triplicate, with standard primers (*Bacteroides*/*Prevotella*/*Porphyromonas* spp. F: GGT-GTCGGCTTAAGTGCCAT, R: CGGA(C/T)GTAAGGGCCGTGC; *Bifidobacteria* spp. F: TCGCGTC(C/T)GGTGTGAAAG, R: CCACATCCAGC(A/G)TCCAC; *Clostridium perfringens* group F: ATGCAAGTCGAGCGA(G/T)G, R: TATGCGGTATTAATCT(C/T)CCTTT; *Lactobacillus* spp. F: CGATGAGTGCTAGGTGTTGGA, R: CAAGATGTCAAGACCTG-GTAAG), exactly as described previously (*Paturi et al., 2010*).

**Table 1** Body weight, food intake, caecal organic acids measured by gas chromatography with flame ionisation detection, and caecal bacterial populations measured by real time quantitative PCR, from C57BL/6 mice fed control or manuka honey-supplemented diets.

| Parameter[a] | Control diet | Manuka honey diet |
|---|---|---|
| Body weight (g) | 24.74 ± 0.35 | 24.77 ± 0.30 |
| Food intake (g/day) | 3.65 ± 0.03 | 3.67 ± 0.03 |
| Caecal short chain fatty acid concentrations ($\mu$mol/g wet weight caecal digesta) | | |
| Formate | 0.45 ± 0.31 | 2.20 ± 1.01 * |
| Acetate | 11.60 ± 0.81 | 11.61 ± 0.75 |
| Propionate | 3.54 ± 0.25 | 3.42 ± 0.26 |
| Butyrate | 2.57 ± 0.35 | 2.92 ± 0.26 |
| Iso-butyrate | 0.18 ± 0.13 | 0.14 ± 0.10 |
| Lactate | 0.49 ± 0.49 | 0.35 ± 0.35 |
| Succinate | 2.13 ± 0.82 | 0.85 ± 0.42 * |
| Caecal microbiota ($\log_{10}$ CFU/g wet weight caecal digesta) | | |
| *Bacteroides-Prevotella-Porphyromonas* group | 10.84 ± 9.96 | 10.80 ± 9.80 |
| *Bifidobacterium* spp. | 9.96 ± 9.39 | 9.73 ± 9.05 |
| *Lactobacillus* spp. | 9.97 ± 9.13 | 9.86 ± 9.02 |
| *Clostridium perfringens* group | 8.80 ± 7.62 | 8.78 ± 7.56 |
| *E.coli* spp. | 7.12 ± 6.62 | 7.13 ± 6.67 |

**Notes.**

[a]Values presented as means ± standard error of the means (SEM)($n = 20$) calculated using Microsoft Excel 2007. Analyses of variance (Genstat Release 8.2, Lawes Agricultural Trust, Rothamsted Experimental Station) were used to calculate the least significant difference (LSD) at $P = 0.05$, with measurement significantly different from control diet marked with *.

## RESULTS AND DISCUSSION

There were no significant ($P < 0.05$) differences in food intake or weight gain data between control and manuka honey-fed animals (Table 1).

We consider changes in the organic acid by-products of microbial fermentation as biomarkers of microbial function (*Rosendale et al., 2011*). Which organic acids are produced depends upon which microbiota members are present, which organic acid pathways they possess and need to use for redox balance, and which substrates they can utilise (*Louis et al., 2007*). Thus, organic acids were measured by GC-FID. There were no significant ($P < 0.05$) changes in the levels of the predominant short chain fatty acids acetate, propionate or butyrate, nor the organic acid lactate or the protein fermentation by-product iso-butyrate (Table 1). There was a significant ($P < 0.05$) increase in formate and decrease in succinate in manuka honey-fed animals, despite high relative inter-animal variation (Table 1). We investigated these acids further. In the absence of stable isotope labels to find the origins of these acids, we chose to look at selected microbial groups which possess the relevant pathways. Formate is almost ubiquitously produced from pyruvate by the gut microbiota. Increased formate concentration may arise from increased production, or from decreased conversion to $CO_2$ or methane (neither measured here). Succinate may be produced and/or utilised as an intermediate from phospho-enol-pyruvate (PEP) or pyruvate via oxaloacetate in a propionate-formation pathway by Bacteroidetes and some Negativicutes (*Reichardt et al., 2014*). The succinate pathway is one of three leading to propionate production.

Given that propionate levels were unchanged, we considered whether changed numbers of the bacteria possessing these different pathways was responsible for changed succinate concentrations. Thus we measured succinate pathway-using caecal Bacteroidetes numbers by RT-qPCR. No significant changes in Bacteroidetes numbers were evident (Table 1). We concede that next generation sequencing-based phylogenetic analysis of the microbiota may have revealed changes within populations, or recorded changes in other groups comprising the microbiota (such as succinate-utilising Negativicutes, or acrylate pathway or propanediol pathway users) which were not considered here. In addition, we belatedly became aware that the *Bacteroidales* family S24-7 do produce propionate and succinate, and are highly (up to 6% in 50% of animals) abundant in the mouse gut (*Ormerod et al., 2016*). Unfortunately, we believe that the primers we used are unlikely to target S24-7 members based on our comparison with the *Bacteroidales* bacterium H10 draft genome (from the uncultured *Bacteroidales* family S24-7 from human gut metagenomic reads; GI: 1050315132, https://www.ncbi.nlm.nih.gov/nuccore/?term=Bacteroidales 20S24-7 accessed 02 November 2016).

We considered the role that MGO may play, should it reach the caecum. Conceivably, exogenous MGO may enter the methyglyoxal shunt to pyruvate, bypassing PEP, glycolysis and the consequent requirement for NAD+ regeneration by formation of lactate or acetate (*Wolfe, 2005*), and pyruvate may in turn be converted directly to formate, consistent with our data. The methylglyoxal shunt is common to gut anaerobes such as bifidobacteria. Additionally, very low levels of prebiotic honey oligosaccharides may have escaped digestion (*Sanz et al., 2005*), and we have previously shown that *Bifidobacteria* responds positively to manuka honey *in vitro* (*Rosendale et al., 2008*). Hence, we measured caecal bifidobacteria by RT-PCR (*Paturi et al., 2010*), but no significant changes in numbers were evident (Table 1).

Finally, for the sake of completeness, we examined other commonly measured caecal microbial groups such as *Lactobacillus*, known to respond to manuka honey (*Rosendale et al., 2008*); *E. coli*, known to be inhibited by manuka honey (*Lu et al., 2013*; *Rosendale et al., 2008*), and the *Clostridium perfringens* group (*Paturi et al., 2010*), representing commensal Gram positive gut microbiota known to be inhibited by manuka honey (*Hammond & Donkor, 2013*), using RT-qPCR (*Paturi et al., 2010*) but no significant changes in numbers were evident (Table 1).

Overall, the effects on gut microbial activity were minimal at the concentration of honey tested. There were some effects on formate and succinate production, but we found no impact on the numbers of some relevant microbial groups, or other less directly-relevant groups. Further investigation with stable isotope labelling, organic acid uptake flux analyses (*Den Besten et al., 2014*) or detailed phylogenetic analyses using next generation sequencing may shed light here, although in light of the current findings we do not consider them to be justified (*Hanage, 2014*).

A final comment relates to dose and delivery of honey constituents to the caecum. This manuka honey delivers ~0.12 mM MGO in the diet, less than half of the 0.3 mM reportedly required to exhibit inhibitory effects against *E. coli* (*Ferguson et al., 1998*). Higher concentrations might be required for the exhibition of antimicrobial effects against the gut microbiota, especially given that losses and dilution would be expected during oral, gastric

and small intestinal digestive processes. However, this feeding study allowed consumption of 2.2 g manuka honey per kg animal weight. Consuming 2.2 g honey per kg bodyweight equates to a 70 kg human consuming at least 150 g of this manuka honey per day, which is a substantial quantity of premium-priced honey; a daily amount unlikely to be achieved, nor sustained for 28 days, by even the most avid manuka honey afficionado.

## ACKNOWLEDGEMENTS

We thank Alison Wallace, Tafadzwa Nyanhanda, Harry Martin and Gunaranjan Paturi of Plant & Food Research, NZ, and Wayne Young of AgResearch, Grasslands, Palmerston North, NZ, for their assistance and advice in the design and completion of the animal feeding trial.

### Funding
D.R. received a PhD stipend under the ''Foods for Helicobacter pylori'' programme funded by the NZ Ministry of Business, Innovation and Employment (Contract No. C02X0402). The programme had some co-investment from Comvita New Zealand Ltd. The funders had no role in study design, data collection and analysis, decision to publish, or preparation of the manuscript.

### Grant Disclosures
The following grant information was disclosed by the authors:
NZ Ministry of Business, Innovation and Employment: C02X0402.
Comvita New Zealand Ltd.

### Competing Interests
The authors declare there are no competing interests.
Juliet Ansell is an employee of Zespri International Limited.

### Author Contributions

- Doug Rosendale conceived and designed the experiments, performed the experiments, analyzed the data, contributed reagents/materials/analysis tools, wrote the paper, prepared figures and/or tables, reviewed drafts of the paper.
- Christine A. Butts conceived and designed the experiments, wrote the paper, reviewed drafts of the paper, obtained ethical approval.
- Cloe Erika de Guzman performed the experiments, analyzed the data, prepared figures and/or tables, reviewed drafts of the paper.
- Ian S. Maddox conceived and designed the experiments, analyzed the data, reviewed drafts of the paper.
- Sheridan Martell and Hannah Dinnan performed the experiments, reviewed drafts of the paper.
- Lynn McIntyre and Margot A. Skinner conceived and designed the experiments, analyzed the data, wrote the paper, reviewed drafts of the paper.

- Juliet Ansell conceived and designed the experiments, analyzed the data, contributed reagents/materials/analysis tools, wrote the paper, reviewed drafts of the paper.

## Animal Ethics

The following information was supplied relating to ethical approvals (i.e., approving body and any reference numbers):

This animal feeding study was approved by the AgResearch Grasslands Animal Ethics Committee, Palmerston North, NZ, Ethics Application 11163.

## Data Availability

The raw data has been supplied as a Supplementary File.

## Supplemental Information

Supplemental information for this article can be found online at http://dx.doi.org/10.7717/peerj.2787#supplemental-information.

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
