# Peer review of "Consumption of antimicrobial manuka honey does not significantly perturb the microbiota in the hind gut of mice"

_PeerJ, doi:10.7717/peerj.2787_

## Round 0.1 · original submission · Minor Revisions

Please revise the manuscript according to the reviewer'scomments.

Reviewer 1 ·

Basic reporting

The paper is well written. The background information, description of methods, results, and discussion of their significance was clearly laid out and easy to follow. Although the primers used were referenced, it would be good to see their sequences listed in manuscript itself.

Experimental design

I have no problem with the experimental design. The high group size used (n=20) is a particular strength.

Validity of the findings

The results obtained support the conclusions drawn in the manuscript, which is that manuka honey, fed at the concentrations described, did not significantly impact the gut microbiota composition using the analysis methods described. It's possible that alternate analyses may tease out some differences, but given the results obtained and current understanding of the composition of manuka honey, any such differences are likely to be subtle. The experiment was adequately controlled and statistically sound.

Additional comments

As far as I'm aware, this study is the first to examine the effects of manuka honey on the gut microbiota in vivo, and as such it contributes useful information to the current body of knowledge for this category of food.

·

Basic reporting

The article is well written and structured.

Submitted supplementary qPCR data is a little confusing. Could this be re-organised? In the data sheets (Bif, Lact/Bact, Clost, E.coli) there are 21 values for each diet (42 individuals), whilst there were only 20 animals reported per group in the text. In the Bact, Clost, E.coli sheet several animals only have data for two targets.

Experimental design

Given the high degree of biological variation between individuals, might a cross over trial have been more valuable? In this case each individual would act as their own control whilst groups could still be compared.

Alternatively relative abundance of bacterial groups could be reported (as would have been reported if a high-throughput amplicon sequencing metataxonomic approach had been taken). This would, of course, require re analysis of all samples using a 'universal' primer set.

The murine gut microbiota often includes relatively large numbers of the Bacteroidales S24-7 family (known to be rich in carbohydrate utilisation genes - see Ormerod et al. Microbiome 2016). Can the authors comment on whether the Bacteroides-Prevotella-Porphyromonas group primers used in this study would also target the S24-7 family members?

Validity of the findings

How is honey formulated into the diet? Is it processed (for example, freeze dried) and if so, does this processing impact on the biological activity of the honey? Given the thrust of the work shows that intake of high levels of manuka honey does not appear to impact on the gut microbiota, it would be important to know that the honey-modified diet is still biologically active.

Could MGO be detected in the caecal material of honey-fed mice (for example by targeted LC-MS analyses)? This information would be valuable when determining the possible impact of the honey diet on the gut microbiota.

---

## Round 0.2 · accepted · Accept

Thank you for the revised version of the manuscript.